# Evaluating the Forest Ecosystem through a Semi-Autonomous Quadruped Robot and a Hexacopter UAV

**DOI:** 10.3390/s22155497

**Published:** 2022-07-23

**Authors:** Moad Idrissi, Ambreen Hussain, Bidushi Barua, Ahmed Osman, Raouf Abozariba, Adel Aneiba, Taufiq Asyhari

**Affiliations:** School of Computing and Digital Technology, Birmingham City University, Birmingham B4 7XG, UK; ambreen.hussain@ieee.org (A.H.); bidushi.barua@bcu.ac.uk (B.B.); ahmed.osman@mail.bcu.ac.uk (A.O.); raouf.abozariba@bcu.ac.uk (R.A.); adel.aneiba@bcu.ac.uk (A.A.); taufiq-a@ieee.org (T.A.)

**Keywords:** forest health indicators, object detection, YOLOv5, WebRTC, 5G, real-time monitoring

## Abstract

Accurate and timely monitoring is imperative to the resilience of forests for economic growth and climate regulation. In the UK, forest management depends on citizen science to perform tedious and time-consuming data collection tasks. In this study, an unmanned aerial vehicle (UAV) equipped with a light sensor and positioning capabilities is deployed to perform aerial surveying and to observe a series of forest health indicators (FHIs) which are inaccessible from the ground. However, many FHIs such as burrows and deadwood can only be observed from under the tree canopy. Hence, we take the initiative of employing a quadruped robot with an integrated camera as well as an external sensing platform (ESP) equipped with light and infrared cameras, computing, communication and power modules to observe these FHIs from the ground. The forest-monitoring time can be extended by reducing computation and conserving energy. Therefore, we analysed different versions of the YOLO object-detection algorithm in terms of accuracy, deployment and usability by the EXP to accomplish an extensive low-latency detection. In addition, we constructed a series of new datasets to train the YOLOv5x and YOLOv5s for recognising FHIs. Our results reveal that YOLOv5s is lightweight and easy to train for FHI detection while performing close to real-time, cost-effective and autonomous forest monitoring.

## 1. Introduction

Forests are an integral part of our environment and hold an essential capacity for the survival of human beings, animals and life in general. Forests and woodlands represent approximately 13% of the total land area in the UK. They provide habitats for myriad animals, insects, etc., and help maintain the balance in the atmosphere. These forest woodland resources are highly valued for a wide range of services, including timber production, water, air-quality improvement, biodiversity and several aesthetic and health benefits for humanity [1]. The monetary value of these UK forests was estimated to be as much as £130 billion [2] as in 2017 and continues to grow. The forests also have a crucial role in combating climate change threats by contributing to urban cooling, mitigation of floods and carbon sequestration. Due to this, it has become necessary to maintain and restore trees and forests through different activities such as seed dispersal, planting trees and preventing trees from falling. In 2019, the program of creating 13,700 Ha of more woodland was initiated in the UK [3], which will be accentuated in the years ahead. Another initiative undertaken in this regard is by the UK’s Woodland Trust, which proposed to plant 50 million native trees over a period of 25 years. Apart from the creation of new forests and conducting surveys, it is necessary to take relevant actions for the conservation work to save the declining species and habitats [4]. Therefore, actions must be taken to protect, conserve and enhance wildlife and fish habitats, and protect other ecosystems, forms of life and natural scenic beauties for tourist and scientific uses. The researchers in [5] reviewed the challenges of major damages to pine and oak trees.

The current practice of managing forests relies on manual data collection, where the forest personnel access the near and remote sites and gather the relevant information. This process of accessing and collecting sufficient data to study the causes of threats to the trees and vegetation to take any corrective measures for the protection of the trees, is tedious and time-consuming. It often involves several visits and in some cases, even can pose risks to the life of forest users. Due to manual collection and handling of data, there is a considerable chance of obtaining erroneous or misinformed data. By coordinating with the forest personnel and other forest users, regular and routine visual tree inspections are carried out, but they tend to be tree-by-tree and depend largely on the enquiries that come in. Their approach of managing forests is usually reactive. Much of their time and financial resources are taken up by survey and maintenance activities to maintain the health of the trees, reduce the risk of trees causing injury to people or damaging the property and respond to public complaints about the tree disservice [6].

Instead of solely relying on manual data collection, in recent years, data collection in UK forests has also involved remote sensing techniques [7]. The data collected by sensors need to be properly assessed [8], but the current methods do not provide an alternative way of validating the data. Data collection concludes with storing the obtained contents in a structured format suitable for further processing. Sensors, Internet of Things (IoT) devices, or cameras can gather data in remote areas of the forest and transmit data to the appropriate storage medium that can be accessed by a server close to the ground station [9]. By providing wireless connectivity between these devices and the server, the manager can monitor these data in a much reduced time frame with less frequent visits to remote areas [10]. This technology has been implemented in several UK forests to monitor tree growth and environmental changes. Such sensor-based systems acquire real-time or close to real-time measurements combined with traditional field studies and long-term records of patterns and processes. This can help the management monitor and respond quickly to environmental changes. Sensor systems such as multispectral cameras, image intensifiers and thermal cameras, whose usage has been previously limited due to the costs or technology considerations, are now becoming widely available and affordable. However, in a lot of these forests, which are declared to be ancient, protected sites, there are several restrictions on safety procedures and operations while deploying sensors or IoT devices with intrusive sensing technologies. Therefore, for future visions of forest monitoring, non-intrusive sensing technologies need to be explored.

In recent years, remote sensing and data processing technologies via UAVs and artificial intelligence (AI) have been incorporated for non-intrusive agroforestry monitoring due to miniaturisation of components and cost-effective sensors. UAVs are enabling new applications for airborne sensing of fires and human or animal activities in forests. Due to their freedom of mobility, UAVs can be used to search for people who lose their way in forests the same way as such systems are used for fast searching for survivors in different dangerous situations such as avalanches [11]. When fitted with suitable sensors, UAVs can be used for rescue missions [12]. For example, thermal cameras [13] fitted to UAVs have helped detect missing persons in the forest, especially during emergencies like fire breakout. The use of low-light visible or new infrared (NIR) cameras can assist in detecting fires in low-light or night conditions [14]. Recent advances in robotics have created opportunities to make things simpler and safer to access information from forests. There are instances where the forest industry uses robots equipped with advanced sensors, computational power, and artificial intelligence to carry out mundane tasks of the forestry personnel, as explained in [15]. In hazardous situations, robotic applications are becoming more popular for decreasing any possible danger to humans [16]. The above state-of-the-art development shows the promises of sensing and robotic technologies in forest monitoring. If developed properly, they can be leveraged for detection and monitoring purposes to help forest managers overcome different challenges and accomplish the work more effectively. In addition to remote sensing technologies, the integration of advanced wireless systems and AI into the forest monitoring system can improve its effectiveness in terms of a faster and more reliable detection of target objects. With the increase of IoT devices and faster communication, the volumes of data generated are increasing, and AI is continuing to integrate for sound and timely decision-making. Continuously evolving computing solutions have been proposed in forest research for data gathering and processing. For forest health assessment and biodiversity conservation, forest management observes multiple key FHIs, including tree species, deadwood, wildlife signs and fire risk. As mentioned earlier, these tasks are performed manually by the forest managers, rangers and volunteers. To accelerate these tasks, computer vision and machine-learning technologies have been leveraged via UAVs in many forestry applications. However, there are many FHIs that can only be observed from the ground, such as wildlife signs, deadwood around the trunk and tree girth and height measurements. There is a limited body of research around using unmanned ground vehicles to detect the above-mentioned FHIs. One of the few studies used a four-wheeled platform equipped with Light Detection and Ranging LiDAR, stereo camera and IMU and a GPS was used to construct a 3D map to evaluate diameter at breast height (DBH) and relative distance between the trees [17]. However, many health indicators such as burrows and deadwood are not reported in this article.

Contributions of this paper can be summarised as follows:We, for the first time, deploy a quadruped robot in a forest landscape and report several vital observations around the robot’s movement dynamics and ability to navigate and survey various forest sites.We define new methods to integrate evolving technologies such as AI, cloud computing, streaming protocols and wireless communication systems to realise end-to-end low-latency detection.We propose a method to maximise the uptime of the monitoring system through load balancing distribution and adaptive offloading mechanisms.We provide analysis of several state-of-the-art object-detection systems in their ability to accurately detect the health of the forest indicators (e.g., tree species, burrows, deadwood, persons and fire) in real time.

The remaining sections of the articles are organised as follows: Section 2 provides an overview of the literature on robotics, AI and ML in forestry. The design of the robotic system, multi-modal sensing, data acquisition and key performance metrics are described in Section 3. The results, challenges and lessons learned are discussed in Section 4, and Section 5 concludes the paper.

## 2. Literature Review

### 2.1. Robotics in Forestry

With the vast development and improvements of technology, unmanned systems are becoming widely popular in the implementations of various use cases. The 21st century has seen a rapid spread of aerial vehicles that do not require control from a human. In some situations, the pilot is nearby; however, in most cases, the pilot is a significant distance away. Bigger drones that weigh over 150 kg are now being used in military applications, while smaller drones below 150 kg have become more popular due to their efficiency in quick manoeuvrability and easy deployment [18]. While their demands are vastly increasing, their manufacturing costs have been reduced by a large margin, resulting in various architectures being developed to meet user needs. With a particular focus on environmental monitoring within rural areas, Vertical Take-off and Landing (VTOL) UAVs are currently considered an ideal vehicle to carry out such analysis in this research [19]. On the one hand, this is particularly useful for analyzing certain forest health indicators (tree canopies and plantations in dense regions) that could be unreachable by humans and ground robots. On the other hand, the agile feature of the VTOL UAVs provides an advantage of quicker orientations to assess certain health factors. However, this can also introduce other drawbacks such as flight policy restrictions, high battery consumption, limited ground view and the forest dynamics creates a challenge for pilots to fly.

With the vast development of AI, sensor technology, computing and vehicle control innovations over the recent years, unmanned ground vehicles (UGVs) have become increasingly popular in both civilian and military applications. This is generally described through two features: (1) the UGV can autonomously drive itself to carry out the missions set by the operator; (2) dependant on the application specifics, the UGV can replace humans in the tasks set [18]. While UGVs are vastly developed and designed in different sizes and structures, there is a lack of forest deployment towards surveying and assessments. For instance, the authors in [20] have developed a wirelessly controlled mini rover that can execute a set of predefined actions automatically. The four-wheeled vehicle has successfully operated with integrated AI sensors to carry out object detection. However, a challenge is introduced in relation to navigating within the natural forest environment. The authors in [21] have also explored forest cleaning through the deployment of a tank-treaded UGV. The AI-powered RGB cameras and LiDAR sensors have successfully classified vegetation compared to waste, giving rise to a challenge related to the environmental condition of changing terrains. One of the robots that was found most suitable to adapt to the changing terrain is the quadruped robot. The authors of [22] advocate for quadruped robots to work in hazardous and inaccessible environments compared to conventional UGVs. Other researchers have looked at the robot’s walking motions using adaptive control algorithms to improve operational lives by enabling smart adaption in varied environmental settings [23,24]. The authors in [25] looked into robotics advancements for forestry applications with a specific emphasis on challenging landscapes. We selected a quadruped robot for forest monitoring since they do not require constant ground contact and can adapt to the changing terrain [9].

### 2.2. AI and Machine Learning in Forestry

Machine learning (ML) is a subset of AI, where a common usage leverages a dataset that is decomposed into training and testing elements for classification and object identification. Introduced in [26], one of the most popular ML algorithms is deep learning (DL), which performs feature extraction from raw data consisting of low-level features to automatically build high-level features. DL algorithms, such as convolution neural networks (CNN) [27] have gained popularity in data analytical research studies, where the learning and classification of large volumes of data are performed effectively. Inspired by the biological neural network, CNN is beneficial for computer vision problems, image classification and object detection. The computation model in CNN is composed of several convolved layers to learn data representation with multiple levels of abstraction [28]. The CNN requires a huge number of annotated samples to estimate millions of parameters, which prevents DL models from being applied to research studies with limited training data. Many CNN-based models have been used in the field of image classification, such as AlexNet [28], ResNet [29] and GoogleNet [30]. For object detection, the CNN-based can be categorised as two-staged and single-staged. Some of the popular two-staged object-detection algorithms are Region-Based CNN (R-CNN) [31], Fast R-CNN [32], Faster R-CNN [33] and Region-Based Fully CNN (R-FCN) [34]. Single-staged object-detection algorithms include Single Shot Detector (SSD) [35] and You Only Look Once (YOLO) [36] series, i.e., YOLOv1, YOLOv2, YOLOv3, YOLOv4 and YOLOv5. Unlike the two-stage detection models, the YOLOv1 detection model has a simple CNN network structure without the extraction process of region proposal. It uses the entire graph as input of the network that outputs the location and category of the bounding box [37]. YOLOv2 uses Darknet-19 for fully convolutional feature extraction and anchor box mechanism, k-means clustering and multi-scale training, which improves recall and accuracy. However, detecting targets with high overlap or small size is still challenging. YOLOv3 adopts a deeper feature extraction network of Darknet-53, multiple scales for prediction, upsampling fusion method and finally, merging three scales, which largely improves the effect of small objects and detection speed [36]. However, the detection accuracy is still not improved, especially when the intersection over union (IoU) > 0.5. SSD combines the concept of regression in the YOLO algorithm and anchor box in Faster R-CNN [35]. YOLOv4 uses CSPDarknet53 as the backbone network and adds weighted residual connection, cross stage partial connection, cross mini batch normalization, self-adversarial training, mish activation, mosaic data augmentation, dropblock and complete intersection over union (CIoU) to the original YOLO framework [37]. In the YOLO series, YOLOv5 is the latest and improved version in terms of speed, size and accuracy.

Computer vision and machine learning have been integrated into many forestry applications. For instance, plant species recognition has been a target of many research studies based on their shoot organ system, such as flowers [38], leaves [39,40], fruit, skin and seeds [41]. Leaves are the most common object of the previous studies because of their distinct shape and structure [42,43] and availability throughout the growing season. Many smartphone-based applications have been developed to facilitate plant recognition, such as ApLeaf [44], leafsnap [45,46], PictureThis [47] and Pl@ntNet [48]. However, these applications depend on the available network connection (limited in remote forests and rural areas) to evaluate the images using trained models on the servers [49]. To overcome this issue, a lightweight model of MobileNetV3 was embedded in an Android mobile application for offline tree species classification [50]. The authors of [51,52,53] focused on tree height and girth measurement using computer vision and image processing techniques.

UAVs have been used increasingly in worldwide forestry applications for image collection and processing. The authors in [54] reviewed drones’ applications in European forests. They concluded that around 36% of the studies focused on estimating the dendrometric parameters, 21% aimed at forest health monitoring [55] and disease mapping [56,57], 14% targeted tree species classification [58], followed by post-fire recovery monitoring and fire measuring (14%) [59,60], quantification of spatial gaps (7%) [61] and the estimation of post-harvest soil displacement (7%) [62]. The sensors used in these UAV-based applications were visible red, green and blue (VIS-RGB), multi-spectral in the visible and near-infrared (VNIR), middle-infrared and thermal infrared (TIR) spectral range, VNIR hyperspectral imaging and LiDAR. For image processing, most applications use the structure from motion (SfM). Many existing works have mapped vegetative status and derived normalised difference vegetation index (NDVI) indexes by recording the spectral range’s reflectance to estimate spatial gaps, post-harvest soil displacement estimation and post-fire monitoring. Most of the UAV-based forestry applications have used Random Forest and support vector machine for tree species surveying and classification [63]. The authors of [64] have used a UAV to obtain the imagery of Mauritia flexuosa (palm tree) and applied DL to segment the images automatically.

## 3. Materials and Methods

### 3.1. Study Area

Figure 1 illustrates the location of where the trial was conducted in the United Kingdom. The ancient woodland is considered to be a royal forest in Nottinghamshire, England, due to its historic features while bordered on the west by the River Erewash and the Forest of East Derbyshire. Its 375 hectares of national nature reserve has been home to hundreds of species including birds, insects, mammals, fungi, trees and plants. Amongst other features, the jewel of this forest is the collection of ancient oaks across the Sherwood landscape area, making it one of the biggest and best locations to detect these species in Europe. Sherwood is thus protected under European law as a Site of Special Scientific Interest (SSSI) and a Special Area of Conservation (SAC), since it is one of the best remaining examples of oak-birch woodland in the UK [65].

### 3.2. Robotic System

#### 3.2.1. Aliengo

As this paper discusses the implementation of a UGV to gather accurate information about the selected environment, these vehicles will be deployed to specifically focus on the understory level of the forest, which is not visible to UAVs from above the tree line. Hence, the Unitree Aliengo robot was selected for this study as the ground robot forest ranger (RFR). Its quadruped formation enables it to achieve multiple motions such as squatting, turning, leaning and moving in various directions, ultimately navigating through rough and unstructured terrains while maintaining stability [66]. The Unitree Aliengo has numerous motions that can be achieved due to its incorporated 12 servo motors, which are placed within positions relevant to the robot fuselage. Hence, 3 degrees of freedom (3-DOF) can be achieved from each leg which totals up to 12-DOF that the robot is able to accomplish. In Figure 2, a 3D model of the Aliengo robot is shown, which consists of numerous coordinate frames. Each coordinate frame indicates that a rotational angle is achievable about a particular axis. It is worth mentioning that the rotational capabilities for all the hip joints are around the x-axis, while the calf and thigh joints are around the y-axis. Four sensors are integrated into the robot foot to detect the contact force between the foot and the ground. With this information, the operator can ensure that the robot maintains a balanced posture in various terrains and operational modes.

In addition to the motion capabilities, the robot is equipped with multiple features that make it suitable for the use case of forest monitoring. The structural developments of the Aliengo are designed using carbon fibre material which delivers high strength with reduced weights. The perception sensors include depth cameras and laser sensors to map the environment in front of the robot. A unique advantage of the Aliengo robot is that it supports external sensors and computers to be integrated with the developed system. The Robotic Operating System (ROS) is used as the main control platform, and utilises artificial intelligence to develop a smart system that can carry out the required monitoring tasks. Additional features of the Aliengo include the following:Reasonable battery life: ranging between 120 and 270 min depending on endurance;Carry payloads of up to 12 kg;Locomotion speeds of up to 6 Km/h;TX2 Jetson for fast computational processes;Mini-Pc for User logic control;Embedded power management system;External manual remote controller;Choice for integrating further tracking, positioning, imaging and communication devices.

#### 3.2.2. Air6 System

To collect relevant data from the forest without causing any harm to the habitat, remote-sensing technologies of UAVs are leveraged for mapping and surveying to assist the forest personnel. The UAVs equipped with camera sensors can capture forest-related data, which can be sent to the ground station through 5G and other wireless networks. Based on these data, faster and more effective decisions can be achieved before taking any necessary actions.

The camera sensors mounted on a UAV act as user equipment (UE) in a mobile network. They can provide a cost-efficient, accurate and flexible solution for surveillance, inspection and delivery [67]. In this case, the UAV must co-exist with ground users and exploit existing infrastructure (such as cellular networks) to transfer collected information to the operator on the ground with certain reliability, throughput and delay, depending on the application requirements [68]. It is expected that 5G and future cellular networks with improved features such as higher capacity, higher reliability and lower latency will be better equipped to deal with the challenges related to networks with UAV [67,69].

For the forest use cases, the UAV chosen for the field trials is referred to as Air6, and is manufactured by Airborne Robotics, as shown in Figure 3. While it carries the VTOL flying capabilities, its hex form enables it to carry larger payloads with higher flight accuracy as compared to the conventional quadrotors [70]. Moreover, numerous programmable sensors which are integrated into the UAV’s control system can provide valuable flight information such as location, altitude, attitude, trajectory tracking and autopilot. The Air6 characteristics are highlighted in Table 1.

#### 3.2.3. Multi-Modal Sensing

The applications of sensors have grown beyond the conventional disciplines of basic sensing due to the vast developments and advancements in micromachinery and easy-to-use microcontroller platforms. Low-cost sensors and easy-to-use devices for monitoring and data collection have significantly grown due to the increasing need for speedy, economic and trustworthy technologies in today’s society. While this research is solely focused on wirelessly sensing of the health of the forest using state-of-the-art technology, multiple health factors which are considered in this study will be assessed through various smart sensors incorporated with AI. Forest rangers generally carry out monitoring and inspections to gather simple data within certain areas in the forest. This will not only consume council resources but will also limit the area that is assessed, resulting in high labour-intensive work, increased costs and time consumption. Therefore, the RFR can be leveraged to support carrying out necessary data collection, which will help maintain the forest ecosystem.

The goal of launching the quadruped RFR in the forest is to enable semi-autonomous roaming and navigation along specific routes while gathering data from various objects, tree species and dead logs using computer vision devices. The RFR will be equipped with high-resolution RGB cameras, laser sensors and wireless connectivity. These are combined to create an intelligent ground vehicle that can support the forest rangers accordingly.

#### 3.2.4. Robot Perception Sensors

As mentioned previously, numerous dynamic motions can be achieved from the robot via the control logic. Advancing this further into autonomy requires completing several steps, which consist of understanding the system architecture in detail and how the operating system interfaces with the actuators to achieve the required objectives.

Figure 4 illustrates the entire control architecture and how the system components are integrated to ultimately control the robot’s motions. For the Aliengo RFR, RS485 can be used as a communication method between the controller and the robot, enabling bi-directional data transfer on the same bus. This method creates a hard real-time computing constraint to guarantee a response within a specified time. As for the two operating systems integrated in the robot, the user logic controller is implemented into the soft real-time computer, which includes high-level control for direct autonomy or low-level control for advanced locomotion controls.

While the controller board acts as the robot’s brain, both of the integrated operating systems work simultaneously together to achieve self-control. It is worth mentioning that low latency is one of the most important aspects when working with autonomy for such robots. Furthermore, precision is of high importance due to the way the robots are structured. As a result, numerous benefits arise from rapid communications, including controller commands and multimedia transmissions, which align with the scope of this research.

The Mini-Pc (Ubuntu 16.04) is the operating system used to achieve direct control of the robot’s motions, while the Nvidia TX2 (Ubuntu 18.04) is used as the multi-modal sensing platform which includes visionary sensing and navigation to mitigate the drawback of slow performance. To improve the environmental monitoring as the robot navigates the forest, an external machine such as Raspberry Pi is used to increase the scope of sensitivity and surveying. For instance, third-party sensing devices can communicate with the robot externally, enabling the robot to improve its autonomy in the deployed field while also collecting additional imperative data.

#### 3.2.5. External Sensing Platform (ESP)

External components can be integrated with the main system to enhance the functionality and operational capabilities of the robot. The platform of the ground RFR has a high payload capacity of 12 Kg, providing it with the ability to carry sensor, computing and communications hardware [71].

Despite the robot being integrated with multiple visionary sensors, the development process has been faced with numerous challenges in relation to autonomy. The machine-learning algorithm was initially applied in the Mini-Pc that was connected to the depth camera; however, a major drawback was introduced in relation to the increased computational efforts, eventually reducing the performance and introducing constant autonomy delays. Moreover, integrated sensors only face forward, which is not particularly suitable for analysing the surrounding environment. Therefore, an ESP carried by the robot is proposed to sense part of the surrounding areas via a GoPro 9 camera, fitted by a gimbal.

Figure 5 illustrates the quadruped robot accommodating the ESP, which consists of the relevant devices and a portable battery. The aim is to collect the information on FHI from trees and process this through ML techniques as the robot moves around the forest. Hence, the set-up incorporates a single-board computer (SBC), RaspberryPi (RPi), which is connected to the GoPro9 camera via the MediaMod interfacing unit. A 73 Wh portable battery is used to deliver long-lasting power to the SBC as it collects the relevant information from the forest.

As for the operational functionalities, the servers running on the cloud computers include a media server and an ML server [72]. The ML server is hosted on the edge computer to perform local computations for object detection using a light version of TensorFlow [73], an open source library to develop and train ML models. The object detection can take place both at the edge computer and at the cloud, switching between them adaptively based on two conditions in a distributed manner. The first condition checks if the network is adequate for data streaming; in the case of high bandwidth, the detection will take place on the cloud. On the other hand, if the network becomes unreliable, the detection will take place on the edge computer. The second condition checks if the battery level on the SBC drops below 20%, resulting in the detection taking place on the cloud to save the battery. This offloading mechanism allows us to freely balance the processing load between the cloud and the local computing power, enhancing monitoring time and minimising the number of missed image frames.

Figure 6 elaborates on the proposed architecture where the media server frontend comprises JavaScript, HTML and CSS, while the backend hosts Node.js with Express. The primary function of the media server is to enable WebRTC communication and create a user-friendly interface. The machine-learning algorithm uses a Flask server to obtain images from the media server, followed by YOLOv5 to detect objects in the image. The outputs are then sent back to the media server via HTTP Post and displayed to the viewer in the form of contours around each object detected. On the other hand, the user views the camera on the Raspberry Pi remotely through a Web page hosted on a Media server while it is connected through a built-in Wi-Fi card. The media server also hosts interactive pages which can display real-time object-detection streams. Appropriate commands can also be sent to the machine-learning server through the media server when required. For example, end-users can set the threshold and enable or disable object detection.

#### 3.2.6. UAV Sensors

While the UAV is flying, multiple sensors are also carried for remote surveying, including a 4 K a6000 24.3 MP CMOS and an FLIR Vue Pro thermal camera. The RGB camera mounted on the UAV can operate in the visible light wavelengths ranging from 400 nm to 700 nm to provide a high-resolution image [74]. The thermal sensor is leveraged for forest health data collection and processing to detect variations in the physical parameters that correspond to a change in temperature [75]. The images captured from the thermal camera generally provide a user-friendly indication based on infrared radiations emitted from objects. These are known to detect radiation in the long-infrared range of the spectrum, roughly between 9000 and 14,000 nanometers. Hence, the operational purpose of this camera is surveying and detecting living beings such as lost persons and animals, or assessing any criminal activities.

### 3.3. Data Acquisition

In this paper, a single-stage object-detection algorithm based on a regression-based algorithm such as YOLOv5, is compared and analysed for each forest use case, i.e., detection of tree species, burrows, deadwood, person and fire through the UAV and UGV. For each use case, a custom dataset is created using mostly RGB cameras. The original size of the images is 4000 × 3000; however, training the YOLOv5 algorithm on the original image increases the computational efforts, which occupies a lot of memory, affecting the responsivity of the algorithm. Therefore, all the images were resized to 640 × 640 pixels and labelled with object classes according to the dataset. These custom datasets are used to train the YOLOv5 model, which is pre-trained on an immense repository containing 330 K images, out of which over 200 K are labelled pictures of 80 different object classes and 1.5 million instances [76]. Among these, more than 800 K instances are person images taken during daylight hours. Therefore, the pre-trained model is used for person detection from the robot. The hardware to train the model included a Lenovo laptop equipped with an 8265U CPU at 1.80 GHz of Intel Core i5 and 8 GB of RAM running on a Windows 10 64-bit system. All the datasets were divided into a ratio of 80:20 to obtain a training set and a validation set, respectively. The hyperparameters for the object-detection model were set as follows: the initial learning rate was 0.01 with decay 0.0005 and momentum was 0.937; the IoU training threshold was set to 0.20.

As for the tree species identification, the dataset consisted of about 300 images labelled with three classes: oak, wood and grass and augmented with variations of horizontal/vertical flip and brightness. The model was trained in 200 epochs which took 37 min approximately. For deadwood, burrows and pine, the dataset included more than 3 K images resized to 416 × 416, augmented with variations of horizontal/vertical flip, rotation, grayscale, saturation and brightness labelled with instances of 6 K for all the classes. The model took 73 min to train in 149 epochs for deadwood and burrows. For fire detection, a publicly available dataset FireNet [77] containing 412 images was obtained. The model was trained on these images with two classes: fire and non-fire. The training took approximately two hours for 1000 epochs. For person detection from drones, the pre-trained model could not be used due to the vertical angle. Therefore, the model was re-trained on a publicly available VisDrone [78] dataset consisting of 288 video clips with 261,908 frames and 10,209 static images. These images are captured by different drone-mounted cameras and cover objects such as pedestrians, vehicles and bicycles. The model took six hours to train in 257 epochs. For person detection in thermal images, the model was trained on 100 images taken from thermal camera. The model took 20 min to become trained in 200 epochs.

### 3.4. Key Performance Metrics

The key performance metrics are identified in this section, highlighting the experiments conducted in the lab followed by field trials to assess the impact weight for all the forest use cases.

Detection Accuracy: In an object-detection model, two crucial objectives are considered, which are classification and localisation. Classification refers to identifying the object and its class, while localisation inspects the coordinates of the bounding boxes around the object. For measuring the performance of an object-detection model, both of these methods need to be evaluated. We describe the evaluation parameters below.

Intersection over Union (IoU): The concept of Intersection over Union (*IoU*) [79] is an essential measure of object detection. It is the intersection over the union of the two bounding boxes; the bounding box for the ground truth (*A*) and the predicted bounding box (*B*). An *IoU* of 1 implies that the predicted and the ground-truth bounding boxes perfectly overlap. If the *IoU* is between 0.5 and 1, the object detection can be classified as true positive. The calculation of IoU is given in Equation (Equation 1): (1)$$IoU=\frac{Area\;of\;Intersection}{Area\;of\;Union}\;\mathrm{or}\;IoU=\frac{A\cap B}{A\cup B}$$

Precision, Recall and Mean Average Precision (*mAP*): *Precision* [80] measures how accurate the predictions are, i.e., the percentage of correct predictions. It is the ratio of the number of true positives (*TPs*), i.e., the number of correctly identified target objects to the total number of positive predictions as shown in Equation (Equation 2). False positive (*FP*) is the number of misidentified backgrounds as the target object.
(2)$$Precision=\frac{TP}{TP+FP}$$

*Recall* [80] is the ratio (Equation (Equation 3)) of the number of true positives to the total number of actual (relevant) objects. False negative (*FN*) represents the number of unidentified object targets.
(3)$$Recall=\frac{TP}{TP+FN}$$

The *mAP* [80] compares the ground-truth bounding box to the detected box and returns a score. The higher the score, the more accurate the model is in its detections. The *mAP* can be calculated by Equation (Equation 4).
(4)$$mAP=\frac{1}{\left|classes\right|}\sum_{c\epsilon classes}\frac{|TP_{c}|}{|FP_{c}|\;+\;|TP_{c}|}$$

## 4. Results and Discussion

### 4.1. Locomotion

Controlling the robot autonomously requires the user to have an in-depth understanding of the robot operating system (ROS). It will always act as the master node of overlooking the robot’s control, including multi-modal sensing. Although this approach is well used in robotics, low latency is significantly important in this research due to the locomotion behaviour and the changing terrain in the forest. Hence, the user datagram protocol (UDP) is considered in this study which has a communication protocol to establish loss-tolerating connections between the integrated devices on the robot. This technique is leveraged to develop an adaptive system that is responsive to this particular use case. Additionally, external sensors such as LiDARs can be incorporated into the ROS algorithm such that the surrounding environment can be fully viewed. As a result, the robot can autonomously move around safely, provided that the perception sensors, as well as the user logic control system, are fully operational.

To demonstrate these commands in a graphical form, Figure 7 illustrates the data collected in terms of the robot’s positional displacement and the body-height adjustments. Figure 7a illustrates the positional displacement when a forward velocity is applied. It can be seen that at 3 s, a 0.1 m/s velocity was initiated in the positive direction in which the robot continued to successfully move forward for 7 s before halting. Although the forward position was continuously increasing, the displacement between 6 and 8 s was decreased due to the change in terrain, which resulted in a marginal error generated from the applied speed.

Figure 7b considers the same concept as Figure 7a but with the velocity applied in the opposite direction. It can be seen that the final position reached was approximately 45% lower than the forward speed. This is due to the structural nature of the Aliengo, which enables it to achieve higher forward speeds in comparison to the reverse speeds. Lastly, Figure 7c illustrates the body height measured from the fuselage. The robot was commanded to change height in order to dynamically manoeuvre in the forest. At every 3 s interval, the fuselage position is decreased where it can be observed that the lowest position achieved by the robot is approximately 0.344 m, while the ordinary standing height is 0.41 m. Hence, it can be concluded that the robot can walk under tree limbs below 0.344 m and can also walk over twigs and logs of up to 0.21 m, according to the experimental tests.

To ensure that the health of the forest is monitored accordingly, the data to be collected must be coupled with its location. This is to help the forest authorities become aware of any negative indicators which will initiate solutions to the problem. Additionally, autonomy is a vital aspect when carrying out monitoring as this will reduce the time and effort of the forest rangers. Hence, collecting such data while having an awareness of the robot’s location will sufficiently complete the assessment criteria for digital monitoring.

According to Figure 8, the robot is equipped with multiple external devices to carry out the required tasks of enabling data transmission with its locations. As the battery life of the robot is limited, a low-cost, lightweight single-board computer is used as an interface device between the data collected from the camera and the GPS location. An Antenna is also used to improve the location accuracy of the robot, which will provide support in quickly detecting the unhealthy factors specified by the forest rangers. With a 5G site prepared by the Nottinghamshire County Council, the 5G dongle was used to interface the connectivity from the 5G base to the SBC, which resulted in a download speed of over 200 Mbps in a rural area.

Regarding technical development, the Sparkfun GPS-RTK2 selected for this research has shown a horizontal positional accuracy of 0.01 m with real-time kinematic precisions and an updated navigation rate of 20 Hz. This device has been directly connected to the SBC, which was set up to collect the longitudinal and latitudinal locations of the GPS. In other words, the real-time geolocation data is consistently collected at every 500 ms interval, followed by directly displaying the data to Google Maps.

As can be seen in Figure 9, the geolocation of the robot is presented on Google Maps using the Sparkfun GPS-RTK2 module. The data collected from Sherwood Forest illustrates the full trajectory of the robot, during which the location changes as the robot surveys the forest. It is worth mentioning that the trajectory data collected from the GPS highlights a constant change in direction. This is due to the dense plantations, trees and large objects within the forest that resulted in the robot safely manoeuvring as the mission is carried out. Additionally, the footprints collected from the robot are achieved from the 5G connectivity covered within the site of the trial. Hence, the collected location had a high accuracy below 0.09 m in comparison to the actual location of the robot. Although the quadruped robot is popular for its adaptability over various terrains, it is worth mentioning that the RFR was operating on a flat surface that had some inclinations between 0 and 5 degrees. The operations conducted during the trial were focused on concrete and grass surfaces below 20 cm. Certain areas consisted of gravel, twigs, thin branches and stones which did not affect the robot’s operational performance.

### 4.2. 4G/Wi-Fi Connectivity for Data Transfer

The detection time, i.e., the time between image data transferred to the cloud and back to the media server via both Wi-Fi (802.11 ac) and 4G is also measured. Figure 10 illustrates the processed frames which took less than 0.5 s when streamed back to the client after detecting objects on the Wi-Fi network. Whereas 50% of the data took up to 0.5 s using the 4G network, while the remaining data took up to 1 s. This shows that wireless network connections can play a pivotal role when transferring data to remote cloud servers for object detection. Technologies that offer higher speed and low latency, such as 5G, could provide faster detection times [72].

Following the load balancing and offloading strategy presented in Section 3.2.5, the time taken to switch object detection from the edge computer to the cloud and vice versa is based on two conditions and is also recorded. Figure 11 shows that approximately 70% of the tests took 2 s to switch from the edge to the cloud computer and vice versa. However, during the trials and experiments, it was determined that the switching from the cloud to the local server occurred at a faster pace than anticipated. The latency in changing from the cloud to a local server is likely due to the severely limited computing resources available on the Raspberry Pi [81], which can be improved using a more powerful SBC such as the Nvidia Jetson [72].

### 4.3. Object Detection in Forest

YOLOv5 is the latest and improved version of YOLO architecture and the first in the series, with the backbone network comprised of PyTorch in place of Darknet. The YOLOv5 series consists of four different versions, i.e., YOLOv5s, YOLOv5m, YOLOv5l and YOLOv5x. These versions differ in the amount of feature extraction modules and convolution kernel location in the network. In turn, the size of the model and the number of parameters also differ in all versions. The input in YOLOv5 uses adaptive anchor frame calculation that adaptively gives the optimal anchor frame in different training sets. The Backbone contains a focus structure to realise the slicing operation, while the Neck uses a new FPN structure to enhance the propagation of low-level features [82]. As a result, YOLOv5 achieves a reduction in computation complexity at least by a factor of four [83].

According to the authors in [84], the YOLOv4 outperforms YOLOv5x; however, the YOLOv5s outperforms YOLOv4 Tiny in terms of average precision (AP). Another comparison is performed by [85] which shows that YOLOv5l outperforms YOLOv4 and YOLOv3 in terms of accuracy of detection. Hence, in this study, we have used YOLOv5s which is a lightweight algorithm with 7.2 million parameters and 270 layers and is easier to use and train. It infers quickly with the fastest detection of 140 frames per second [86] and performs better than the previous versions. The size of the weight file of YOLOv5s is 90% less than YOLOv4, indicating its suitability for deployment in the embedded devices to implement real-time detection [87,88], performing less computation to save battery power. Therefore, for time-critical applications such as the detection of fire or a person lost in the forest, YOLOv5s has the potential to be most effective for object-detection operations. The architecture of the YOLOv5 and detection of oak, grass, fire and deadwood are shown in Figure 12.

We trained YOLOv5x and YOLOv5s models on our datasets to compare the results and size of the weight file shown in Table 2. It can be noticed that the size of the weight file with YOLOv5s is only 14.5 MB, whereas with YOLOv5x it was 173 MB. According to [89], the size of the model with YOLOv4 was 244 MB and with YOLOv4-Scaled it was 401 MB, which is quite huge for computation on an SBC like Raspberry Pi. The tree species detection from the ground is difficult because of varied leaf orientation, colour and distance from the robot. Therefore, for the tree species identification, i.e., pine and oak, the detection accuracy was 53% and 10% with YOLOv5x and 55% and 13%, respectively, with YOLOv5s. For deadwood and burrows, the detection was with an accuracy of up to 64% with YOLOv5x and up to 67% with YOLOv5s. As for the fire, both models could detect the fire instances with an accuracy of up to 75%. For person detection from the drone, the models could detect pedestrians with up to 43% accuracy. The person could be detected in thermal images with an accuracy of 30%. After the models were trained on each dataset, unseen pictures and videos were fed for the model inference with a confidence threshold of 0.25. The accuracy can be improved by training the algorithm on more pictures and epochs. The person detection in thermal image from the drone is shown in Figure 13. The results of FHI detection are shown in graphs in Figure 14.

### 4.4. Challenges

#### 4.4.1. 5G Network Operational Limitations

One of the main challenges faced in setting up the 5G connectivity within the forest is the unavailability of spectra for providing mobile services, which is due to the lack of ownership of spectrum radio sharing. The network was developed from the ground up to support a wide variety of services, devices and deployments; 5G will encompass a diverse spectrum. It is built on established technologies to ensure backward and forward compatibility [90]. The 5G base station (BS) in Sherwood Forest provided 5G coverage extending up to a sector of a sphere with a radius of approximately 200 m. To avoid interference due to signal penetration from trees, it was necessary to have a strong, short-distance line of sight (LOS) transmission link between the BS and the UAV/UGV. With the current specifications of the antenna height at around 10 m and the surrounding trees of comparable height to that of the antenna, it was technically challenging to form an LOS transmission link between the BS and the UAV which flew above the tree canopies. Hence, an alternative approach was proposed for the UAV monitoring, which consisted of capturing angular images of the trees at a lower altitude. The density of plantations and trees in the forest makes it difficult for conventional UAVs to safely manoeuvre the forest.

#### 4.4.2. UAV and UGV Constraints

From the given regulations of flying a drone for surveying, the maximum altitude to be flown must not exceed 50 m for safety purposes. With this constraint, the remote sensing activities affected the resolution of the data captured. Moreover, the maximum thrusts generated by the UAV actuators sum to approximately 20 Kg, which generates a restriction on carrying multiple sensors in a single flight. Before the experiments were carried out, certain rules were proposed by the forest management to operate the drone within a forestry site. These were: (1) drones cannot be flown over the site to gather the HoF data during the breeding season (i.e., March–September) every year; (2) the drone can be flown to collect tree-health data outside this time under a general film agreement in areas approved by the forest authorities; (3) the drone will need to be flown by a licensed drone pilot, who will be responsible for alerting the Civil Aviation Authority (CAA) and following their regulations; (4) the deployment of ground robots is to remotely support forest rangers with the tasks on the ground.

Robots are programmable machines that can aid forest personnel when performing tasks that are (a) repetitive, (b) expensive and ineffective when the benefit is not sufficiently large compared to the cost and/or (c) dangerous and life-threatening [91]. However, using robots in forests brings a new set of challenges. For instance, it is challenging to design a suitable programmable machine that can work in a hostile forest environment as the terrain is uneven and unpredictable, with many physical obstacles and the remoteness of the forest sites. The significant issues that need to be addressed when using robots therein are localisation, navigation and mobility. Moreover, to conduct the tests in the field with robots and drones, the weather conditions must be suitable for their operation. For instance, a wet and rainy environment can generate instability, which could damage the mechanism and its sensors. Therefore, such field tests have to be conducted on a dry day [92].

With the unconditional rules and restrictions of flying drones as imposed by the UK CAA and forest authorities, the policy of deploying RFRs can be less cumbersome due to the naturally reduced risk as compared to the UAV. With this being said, the RFR can manage to collect the data, but the deployment of a UAV will provide larger area coverage and further data collection. For instance, the RFR can be utilised to cover the ground area such as the detection of burrows while the drone can cover areas that are challenging to observe through the RFR.

#### 4.4.3. Data Streaming and Object Detection

While implementing a real-time object-detection application, the main challenge is identifying a low latency streaming protocol. The reason was that many online resources and libraries are streaming protocols with high latency that could not be used for real-time object-detection applications. For example, the Real-Time Streaming Protocol (RTSP) and HTTP Live Streaming (HLS) [93,94], which are widely used for streaming, offer latency of up to 3 s. This was more than what was required for time-critical use cases. We use Web Real-Time Communication (WebRTC) [95], which provided a latency that was less than 250 ms. However, it also presented a new challenge, i.e., WebRTC was not supported in Python. This is because WebRTC only works in the browser and cannot be accessible in Python via a URL link like the other streaming protocols (e.g., RTMP and HLS).

To overcome this challenge, instead of using WebRTC directly in Python, the frames were sent from WebRTC to Python over HTTP Post [96] and the results were returned to the browser via HTTP Get. For example, the getUserMedia() [97] method was used to access the user’s camera through the browser directly. The image was captured from the camera using a canvas [98]. The HTMLCanvasElement.toBlob [99] method was used to convert canvas images into a blob. After the conversion, HTTP Post was used to send data to the YOLOv5. From there, the output was wrapped into a JSON object [100] and sent back to the browser. Once the JSON response was received from YOLOv5, another canvas was created that drew boxes around the detected objects using the outputs. This provided a way to use WebRTC in Python object-detection frameworks without using it directly in Python.

Other challenges with WebRTC included testing its latency when applied to real-time applications. Although WebRTC provides tools for testing latency (e.g., chrome webrtc-internals), these tools use Round Trip Time (RTT). The latter and latency are similar but not identical. For example, RTT is the time it takes for a packet to be sent to a destination plus the time it takes for an acknowledgement of that packet to be received back at the origin. In contrast, latency is only the time it takes for a packet to be sent to a destination. Furthermore, it cannot be assumed that latency is equal to half of RTT because delay can be unequal between any two given endpoints.

#### 4.4.4. Data Limitations

Limited datasets introduce the problem of overfitting. Therefore, to acquire good results of accuracy in detecting FHI, machine-learning methods require a substantial amount of data. Currently, available datasets are either insufficient or do not contain images of indigenous tree species and certain FHI, such as wildlife signs and deadwood. A complete dataset must contain images captured from different angles under different conditions as much as possible. Gathering images for a custom dataset is a time-consuming task, which requires expert knowledge for labelling images.

One major problem with a custom dataset is class imbalance [101], i.e., the unavailability of a balanced number of instances per available class. This can result in a high rate of false negative indicators. The problem of imbalanced data was handled by preparing the custom dataset for each FHI with 3K images and instances using the GoPro9. To increase the size of the dataset, each dataset was augmented with different variations such as horizontal/vertical flip, rotation, grayscale, saturation and brightness. For deadwood and burrows, the images were also captured under low-light illumination at around the sunset. This helps to detect FHI in dense forest areas that have a limited flow of natural light.

One of the main challenges is the variation in seasons, which impacts the colour and the shape of leaves. As a result, some of the machine-learning algorithms designed to detect objects that are found in nature at certain seasons only cannot be used for testing and validation. For example, oak tree leaves are not available in winter for the application of the machine-learning algorithm to accomplish species detection. However, monitoring the trees during leaf season can provide indications of tree health. In addition, burrows and deadwood detection can be achieved all year round—an important indicator of several tree species’ health such as oak.

### 4.5. Future Works

Following the successful development of a surveying system that can observe and analyse the forest’s health using UAVs and UGVs, several approaches and techniques have been considered towards improving the current performance of the developed system.

This research mainly focused on utilising the 4G/Wi-Fi connectivity to transfer data. However, parts of the forest area had no connectivity, which resulted in the robotic vehicle losing relevant data. Hence, an alternative approach was considered which consists of continuing the robot’s operation alongside a high processing computational machine as a data aggregator to offload the data collected once a stable connection is acquired. This will not only provide further information about the forest but will also maintain the consistency of surveying various sites. Moreover, detecting the location of certain health factors once the robot completes its mission was found challenging due to the dense plantations in the area. For instance, the UGV which is equipped with an accurate GPS module only provides the location of the robot and not the exact position of where the health indicator is positioned. Therefore, future works can be considered in determining the exact location of the plantation rather than the robot itself.

Many valuable lessons are learnt while implementing the WebRTC application and machine-learning tools. For example, to understand the core concepts of WebRTC, the WebRTC signalling server was explored with a consideration of how data connections are built between clients and how they are shared through the signalling server. As the majority of WebRTC applications are written in JavaScript, new JavaScript APIs are now being further explored while developing real-time object-detection applications. Hence, future works will emphasise utilising the new JavaScript APIs to generate an optimized algorithm concerning operating the WebRTC whilst ensuring that the streaming is secured. As for the machine-learning algorithm, carefully collecting and annotating training images can significantly improve the final model’s accuracy. However, training the algorithm tends to be time-consuming depending on the images to be trained. Hence, configuring external graphical processing units (GPUs) can support training models faster, which would efficiently result in the algorithm performing at higher accuracy based on additional training images. Further methods such as feature extraction will be explored to improve the accuracy of tree species detection from the ground. We include the usage of 5G networks and more powerful SBC such as NVidia Jetson for real-time data transfer and object detection in future works.

## 5. Conclusions

In this article, we proposed a smart digital monitoring system operated on UAVs and UGVs to identify certain health factors that contributes to deforestation, person and fire. Although manual data collections of certain forestry sites are still the conventional approach, the experimental deployment of UAVs and UGVs with the incorporation of AI and wireless connectivity has provided sufficient information concerning the surveyed sites. An Air6 Hexacopter was leveraged to carry out remote sensing of tree species, tree canopies, fires and persons while the quadruped robot was able to successfully observe and survey the bottom layer of the forest including, dead logs, burrows and persons. With this being said, both vehicles were equipped with visionary sensors to carry out various object defections while feeding this data through the 5G-connected WebRTC platform. As for the quadruped robot, multiple ground experiments were carried out in the rural area of Sherwood Forest, which includes analysing the robots’ motions and capabilities with the uneven terrain. It was observed through the GPS module operated on the SBC that the robot was able to successfully manoeuvre the forest on grass and footpaths. The object-detection algorithms such as YOLOv5x and YOLOv5s are compared in terms of their accuracy and size (weights file), concluding that YOLOv5s provides the same accuracy as other algorithms; however, it is much smaller in size as compared to YOLOV4, YOLOv4-Scaled and YOLOv4-Tiny which makes it the right candidate to be deployed on embedded devices with less computation capabilities and limited battery for prolonged forest monitoring. The incorporation of the detection algorithm provided a good accuracy concerning all the objects, which indicates that the proposed Hexacopter and quadruped robot can sufficiently support environmental monitoring and surveying.

## Figures and Tables

**Figure 1 sensors-22-05497-f001:**
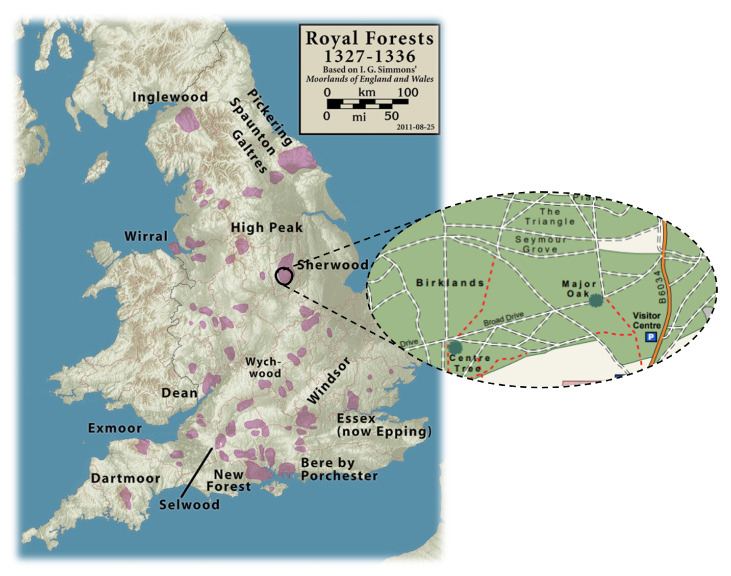
The location of Sherwood highlighted in the UK.

**Figure 2 sensors-22-05497-f002:**
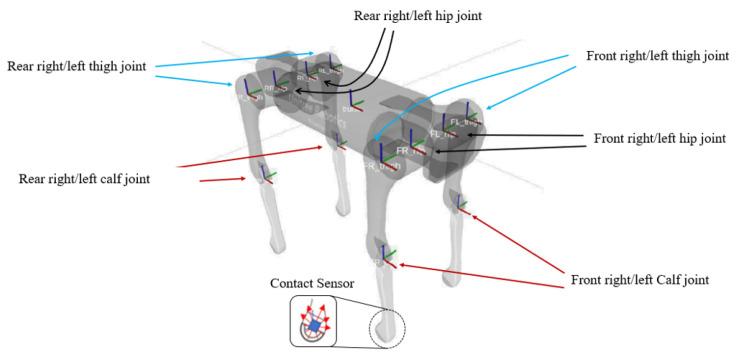
Body joints on the quadruped robots to achieve locomotion.

**Figure 3 sensors-22-05497-f003:**
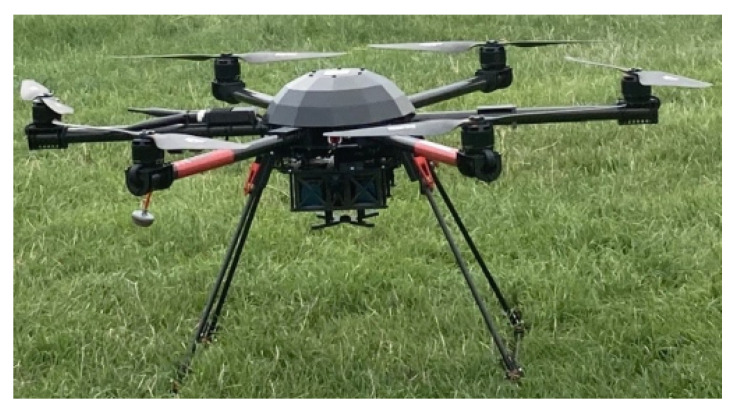
Airborne robotics modelled Air6 UAV copter.

**Figure 4 sensors-22-05497-f004:**
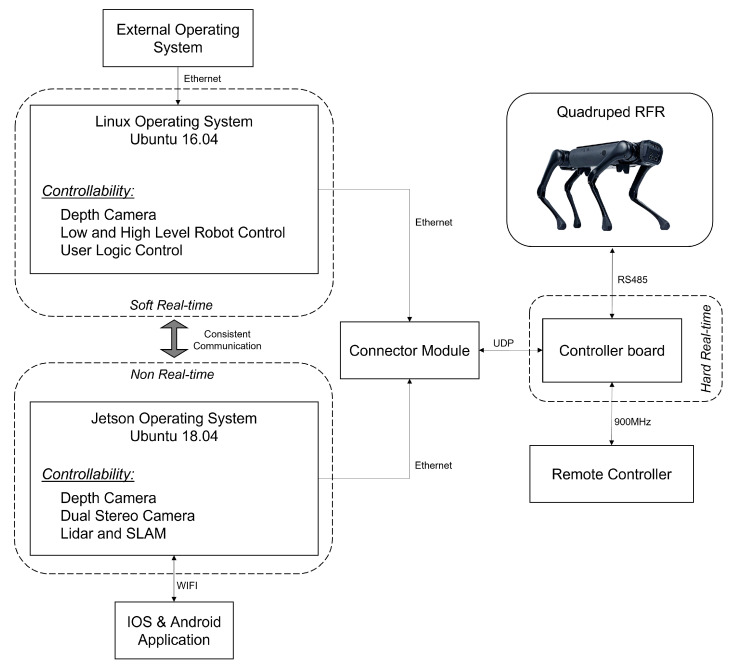
Full system architecture of the Aliengo robot.

**Figure 5 sensors-22-05497-f005:**
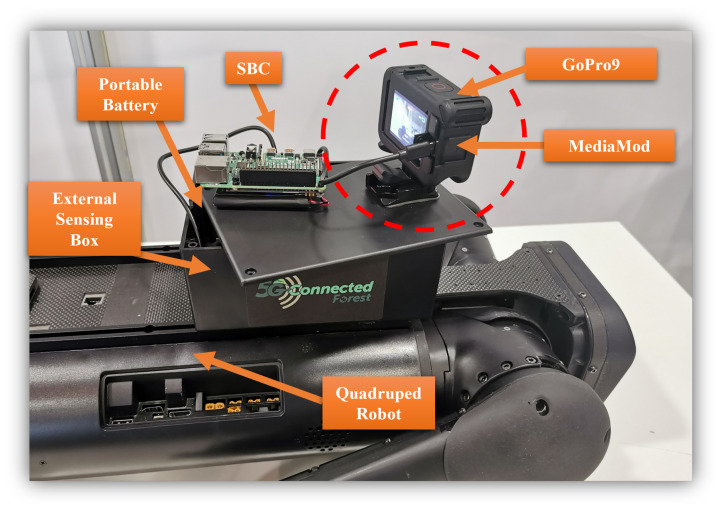
External visionary sensor carried by the Aliengo to monitor the forestry environment.

**Figure 6 sensors-22-05497-f006:**
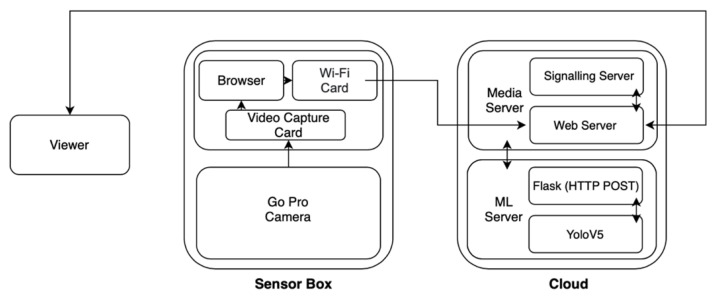
System architecture for the external sensor box carried by the Aliengo.

**Figure 7 sensors-22-05497-f007:**
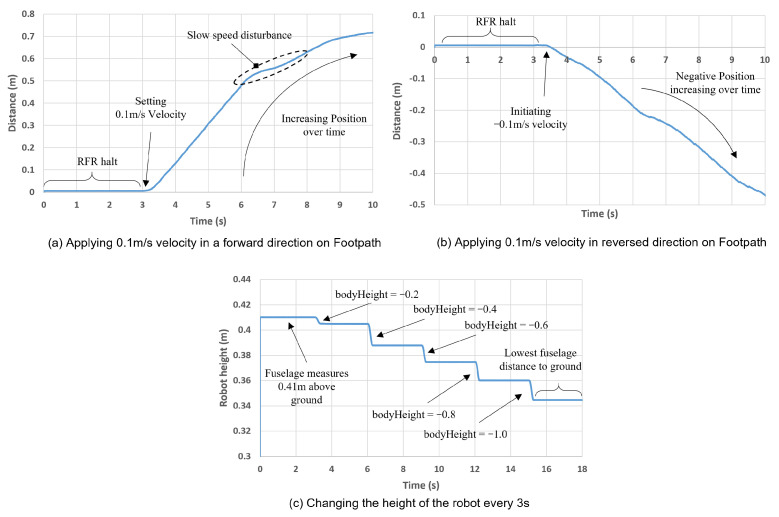
Measuring the positional displacement and the body height to dynamically manoeuvre in the forest.

**Figure 8 sensors-22-05497-f008:**
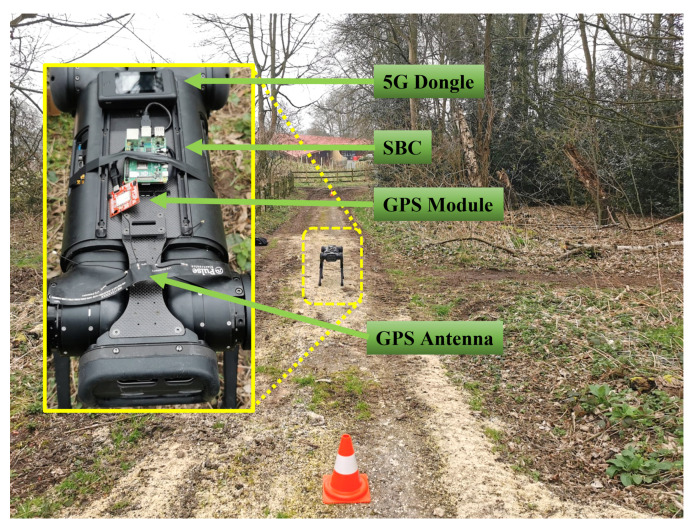
Quadruped robot equipped with a GPS module connected to 5G.

**Figure 9 sensors-22-05497-f009:**
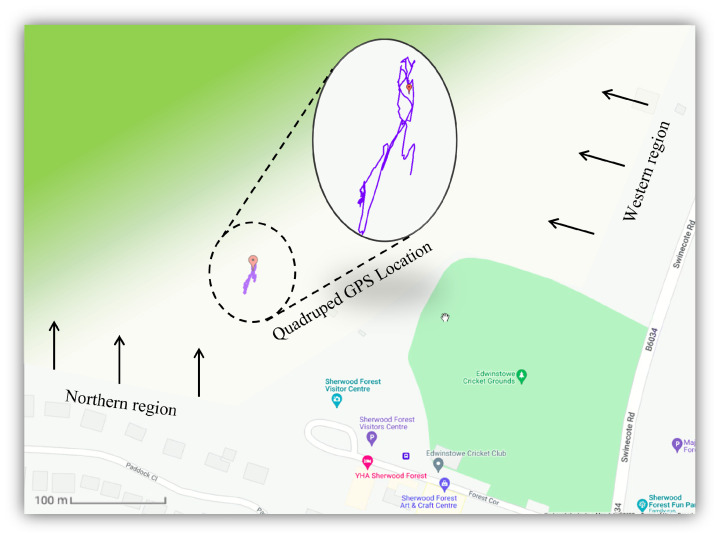
Google Maps demonstrating the tracked trajectory of the quadruped robot in real-time.

**Figure 10 sensors-22-05497-f010:**
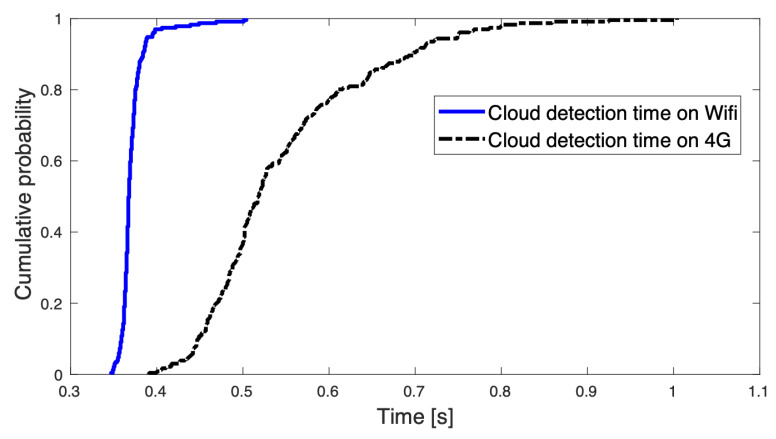
Cloud detection time on 4G and Wi-Fi [72].

**Figure 11 sensors-22-05497-f011:**
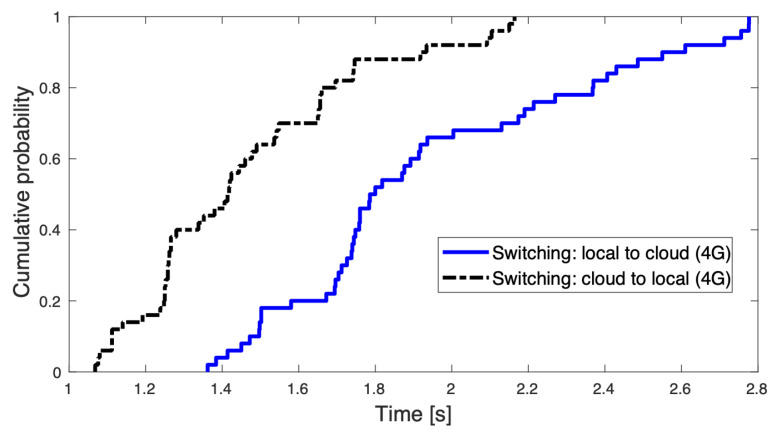
Switching time between edge and cloud computing on 4G network [72].

**Figure 12 sensors-22-05497-f012:**
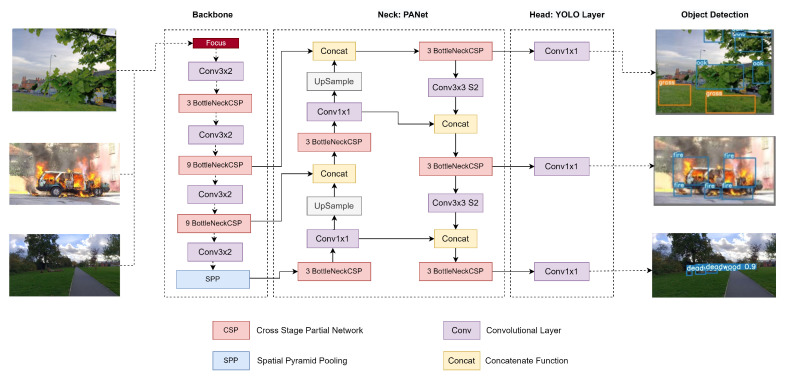
The architecture of YOLOv5.

**Figure 13 sensors-22-05497-f013:**
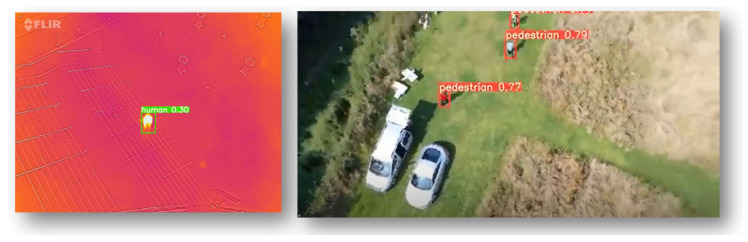
The person detection from drone in thermal and RGB images.

**Figure 14 sensors-22-05497-f014:**
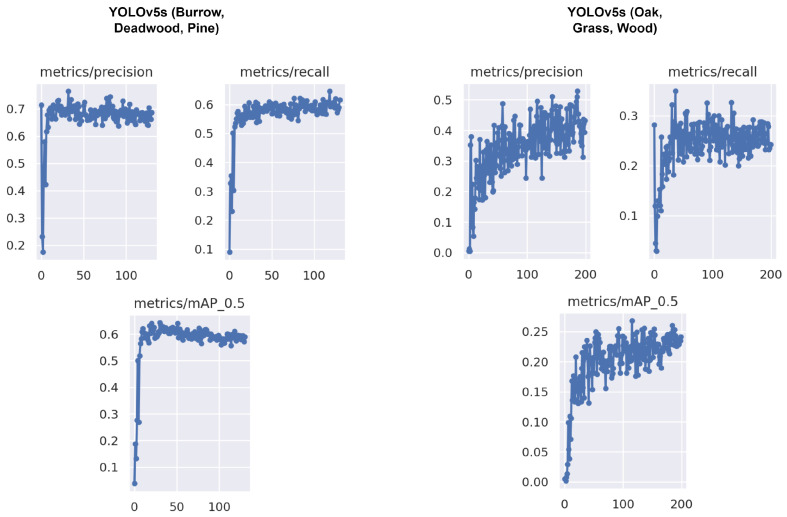
The result graphs of YOLOvs on FHI detection.

**Table 1 sensors-22-05497-t001:** Air6 Characteristics.

Components	Features	Remarks
Motor Power	6 × 750 W	Brushless
Weight	4 kg	Without Payload
Maximum take-off weight	6.5 kg	-
Flight-Controller	PixHawk	Redundant Controller
LiPo Battery Pack	2 × LiPo 3700 mAh 6S	1 Flight Pack = 2 pcs Battery
RC Remote Control	204 GHz	Range approx. ca. 1.500 m (no obstacle)
FPV Video-uplink	Analog 5.8 GHz	Digital 5.8 GHz
Gimbal (Camera Mount)	2–3 axis-brushless	Remotely operated via remote control
Maximum Wid Speed	10 m/s	-

**Table 2 sensors-22-05497-t002:** Results of comparing YOLOv5x and YOLOv5s models for FHI, Fire and Person detection.

Algorithm	YOLOv5x			YOLOv5s		
Size	173 MB			14.5 MB		
Classes	Precision	Recall	mAP@50	Precision	Recall	mAP@50
Burrow	70	66	64	79	64	67
Deadwood	68	60	62	67	56	58
Pine	63	52	53	63	62	55
Grass	88	30	40	64	30	35
Oak	39	0.09	10	28	17	13
Wood	39	30	23	36	38	27
Fire	85	65	75	81	74	71
Pedestrian	65	35	40	58	41	43
Person (Thermal)	30	20	28	25	20	30

## Data Availability

The data presented in this study are the property of Birmingham City University and openly available at http://www.open-access.bcu.ac.uk/13391/. Please cite the article should you require to download and use the dataset.

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
