# Peer review of "Evaluating the Forest Ecosystem through a Semi-Autonomous Quadruped Robot and a Hexacopter UAV"

_sensors, 2022, doi:10.3390/s22155497_

Round 1
Reviewer 1 Report
Reviewer’s comments
to the manuscript “Evaluating the Forest Ecosystem through a Semi-Autonomous
Quadruped Robot and a UAV" (Authors: Moad Idrissi, Ambreen Hussain, Bidushi Barua1, Ahmed Osman, Raouf Abozariba, Adel Aneiba and Taufiq Asyhari).
The article is devoted to remote sensing and data processing technologies via Unmanned Aerial Vehicles (UAV) and Artificial Intelligence (AI) for non-intrusive agroforestry monitoring. In the research, UAV is used to observe Forest Health Indicators (FHI) inaccessible from the ground. Also quadruped robot was brought to observe FHI from the ground to reduce the time and effort of forest personnel to carry out various surveying tasks. State-of-the-art sensors, computing, and communication modules are also utilized for data processing and transmission, including integrated sensors and an external sensory platform. Custom datasets are constructed to train the object detection algorithm YOLOv5 for recognizing FHI, fire, and persons. Authors achieved a high accuracy of 99% in FHI detection.
There are some other points to correct or to make the information more exact:
Essential drawbacks.
Remark 1. In “Sensors” (https://doi.org/10.3390/s21030974) a comparison between YOLOv4 and YOLOv5 is made. In that article, the best efficiency was given by YOLOv4. This casts doubt on the result obtained by the authors of this work. Authors should better work on this issue.
Remark 2. Line 535. “YOLOv5 is the latest and improved version of YOLO architecture and the first in the series, with the backbone network comprised of PyTorch in place of Darknet.” However, YOLOv4 also has PyTorch implementation. The backbone feature extractor for YOLOv4 and YOLOv5 is the same. If the PyTorch implementation is more effective, why the authors in the article do not compare the same implementations or at least give a link to a performance comparison?
Remark 3. Line 537. It is not clear from the article which neural network was taken, since there are many modifications of YOLOv5, for example YOLOv5l model has 47 million parameters and 392 layers and YOLOv5x model has 87 million parameters 476 layers, the same for YOLOv4. Also, there is a modification of YOLOv4 – Scaled YOLOv4 (arXiv:2011.08036), which is superior to YOLOv4, why is Scaled YOLOv4 not used initially? The authors should specify which type of neural network was used in the research.
Remark 4. It’s unclear whether the authors have optimized and configured the neural network hyperparameters on the dataset, and here is their best result or not. Are they the best or did it just happen? The authors should analyze it in more detail, and show the formulation of the experiment, its reproducibility.
Technical drawbacks.
Remark 1. Line 118, 464. Correct is LiDAR.

Reviewer 2 Report
The author has conducted an interesting study using a semi-autonomous quadruped robot and UAV. The data is processed and transmitted using sensors, computing and communication modules, and finally the forest ecosystem is assessed. It has good practicality and has some research value. However, there are some doubts and the following modifications are suggested.
1. The authors should give an overview map of the study area to facilitate the reader to better understand the relevant research.
2. It is known that not all forest terrain is flat, and there are many forests with complex terrain conditions, such as streams, shrubs, rocks, etc. Can the machine dog work properly. Does the method have universal applicability?
3. How fast can Quadruped robots acquire data from the whole forest, and is it feasible to collect large scale data in a short time like UAV? Does it affect the overall research progress?
4. The author mentions in line 578“the dynamic nature of the trees in the forest“, please explain it.
5. Whether the authors can give information about the images or other data acquired by UAVs and UGVs, the reader cannot judge the accuracy of both at present.
Round 2
Reviewer 1 Report
The article can be accepted in the submitted form.